# Factors affecting postnatal care service utilization in Pyuthan district: A mixed method study

**Tulsi Ram Thapa**[1,2]*, **Reshu Agrawal Sagtani**[2], **Anita Mahotra**[3], **Ravi Kanta Mishra**[1,4], **Saraswati Sharma**[2], **Sudarshan Paudel**[2]

**1** Policy, Planning and Monitoring Division, Ministry of Health and Population, Kathmandu, Nepal, **2** School of Public Health, Patan Academy of Health Sciences, Lalitpur, Nepal, **3** Faculty of Medicine, Public Health and Nursing, Universities Gadjah Mada, Yogyakarta, Indonesia, **4** Bergen Centre for Ethics and Priority Setting, Department of Global Public Health and Primary Care, University of Bergen, Bergen, Norway

* tulsiram.thapa730@gmail.com

**Data Availability Statement:** All relevant data are within the paper and its Supporting information files.

## Abstract

The first hours, days, and weeks following childbirth are critical for the well-being of both the mother and newborn. Despite this significance, the postnatal period often receives inadequate attention in terms of quality care provision. In Nepal, the utilization of postnatal care (PNC) services remains a challenging issue. Employing a facility-based concurrent triangulation mixed-method approach, this study aimed to identify factors associated with PNC service utilization, as well as its facilitators and barriers. A quantitative survey involved 243 mothers who had given birth in the six months preceding the survey, selected using a multistage sampling technique from six health facilities of two randomly selected local levels of the Pyuthan district. Weighted multivariate logistic regression was employed to identify predictors of PNC service utilization. Additionally, qualitative analysis using Braun and Clarke's six-step thematic analysis elucidated facilitators and barriers. The study revealed a weighted prevalence of PNC service utilization as per protocol at 38.43% (95% CI: 32.48–44.74). Notably, Socioeconomic status (AOR-3.84, 95% CI: 2.40–6.15), place of delivery (AOR-1.86, 95% CI: 1.16–3.00), possessing knowledge of postnatal care (AOR = 6.75, 95% CI: 3.39–13.45) and access to a motorable road (AOR = 6.30, 95% CI: 3.94–10.08) were identified as predictors of PNC service utilization. Triangulation revealed knowledge on PNC, transportation facilities, PNC home visits, and postpartum weaknesses to visit health facility as areas of convergence. Conversely, divergent areas included the proximity of health facilities and the effect of COVID-19. The study identified a low prevalence of PNC service utilization in the district. To enhance utilization, targeted interventions to increase awareness about postnatal care, appropriate revision of existing policies, addressing wider determinants of service utilization, and ensuring effective implementation of PNC home-visit programs are of utmost importance.

**Funding:** The author(s) received no specific funding for this work.

**Competing interests:** The authors declare that they have no competing interests.

## 1. Introduction

Globally an estimated 287,000 women died during and following pregnancy and childbirth, and majority (95%) of these deaths occurred in low and middle-income countries, of which most could have been prevented [1]. Despite highest percentage reduction in maternal mortality rate between 2000 and 2017, South-East Asia Region still has third highest maternal mortality ratio (MMR) among World Health Organization's region [2]. In Nepal, the maternal mortality rate (MMR) is reported to be 151 per hundred thousand live births, with a notable 61% of deaths observed during the postpartum period [3]. The neonatal mortality rate (NMR) stands at 21 per thousand live births, and postnatal check-ups as per protocol, remains at approximately 40% indicating Nepal's off track trajectory towards achieving Sustainable Development Goals (SDGs), which aim to reduce MMR to 70 per hundred thousand live births, neonatal mortality rate to 12 per thousand live births, and increase postnatal care (PNC) check-ups as per protocol to 90% by the year 2030 [4–6].

Globally, 81 Countdown countries account for 95% of maternal and 90% of child deaths, with most of these deaths occurring within 42 days after childbirth. Despite the critical importance of this period for both maternal and child survival, postnatal care consistently has one of the lowest coverage of interventions on the continuum of maternal and childcare. In Countdown countries, the median coverage is reported to be just 36% for postnatal care for babies and 62% for postnatal care for mothers [7]. In Nepal, the Safe Motherhood Program, initiated in 1997, with the aim to mitigate maternal and neonatal mortality and morbidity through a comprehensive continuum of care from the antenatal to the postnatal period. Although notable achievements have been achieved in various facets of Nepal's Safe Motherhood Program, a critical concern persists regarding Postnatal Care (PNC). In Nepal, there is a substantial discrepancy of forty percentage points between the rates of women receiving the four antenatal care (ANC) check-up as per protocol and the three postnatal check-up as per protocol [5]. The utilization of PNC services as per protocol was low in Pyuthan district as compared to other component of safe motherhood program. Only 17.3% of the postpartum mothers had PNC check-up as per protocol in the FY 2018/19, while the figures for four ANC check-up as per protocol, institutional delivery, and first PNC check-up were 51.1%, 63.8%, and 63.5% respectively. More than 46% of the postpartum mothers were missed between the first and third PNC check-up. Considering this low utilization of PNC services as per protocol, PNC Home-Visit Program was implemented in the district since FY 2019/20 [8].

The postnatal period, usually defined as the first six weeks after childbirth, provides an opportunity for various promotive and preventive interventions, such as nutrition education, hygiene and breastfeeding promotion, family planning counseling, immunization, vitamin A supplementation, and more [9–11]. Poor utilization of PNC services not only deprive mothers and babies from curative, promotive and preventive aspects of interventions, but also leads to complications that heighten the risk of deaths along with negative financial and productivity consequences [12]. To address the poor PNC service coverage, centrally planned and blanket approach of the program may not work in country like Nepal where there are large geographical and cultural variations. So, for the contextualized evidence and intervention, it is imperative to identify the context specific facilitators and barriers. Furthermore, There are limited body of published literatures in Nepal that solely focuses on PNC service utilization as per protocol, and those published are more concentrated on identifying barriers [13, 14]. This study aimed to identify both facilitators and barriers of PNC service utilization.

## 2. Materials and method

### 2.1 Study design and study site

Mixed-method designs have been gaining importance in recent years and have emerged as a third research paradigm. The inherent weaknesses of positivism (quantitative method) and interpretivism (qualitative method) can be better addressed by combining both philosophies to address common research questions [15]. Our study employed a concurrent triangulation mixed-method design, allowing us to compare quantitative and qualitative data to see if the data confirm (convergent findings) or disconfirm (divergent findings) each other. The quantitative component focused on assessing PNC service utilization and associated factors, whereas the qualitative component explored facilitators and barriers. Both quantitative and qualitative data were collected simultaneously between September and October 2021.

### 2.2 Study population

For quantitative component, information was collected from postnatal mothers who were between one week to six months of postpartum and whose current child was alive. For qualitative component, in-depth interviews (IDIs) were done with postnatal mothers whereas key-informant interviews (KIIs) were taken with Female Community Health Volunteers (FCHVs), Auxiliary Nurse Midwives (ANMs), Health Coordinators (HC) and Public Health Nurse (PHN) (Table 1).

**Table 1. Participants' details involved in qualitative study.**

| In-depth Interview | | | | | | | |
|---|---|---|---|---|---|---|---|
| S.N. | Code | Age (years) | Ethnicity | Education | Place of residence | No. of PNC check-up | Type and place of delivery |
| 1. | P1 | 25 | Brahmin | Grade 8 | Jhimruk-8 | 3 | Normal, Home |
| 2. | P2 | 32 | Dalit | Illiterate | Jhimruk-7 | 2 | Normal, HP |
| 3. | P3 | 37 | Dalit | Illiterate | Jhimruk-7 | 3 | C/S, Hospital |
| 4. | P4 | 25 | Chhetri | Bachelor | Jhimruk-2 | 3 | Normal, HP |
| 5. | P5 | 22 | Janajati | 10+2 | Jhimruk-6 | 1 | Normal, HP |
| 6. | P6 | 25 | Janajati | SLC | Pyuthan-10 | 1 | Normal, Hospital |
| 7. | P7 | 23 | Janajati | Grade 6 | Pyuthan-6 | 3 | Normal, Hospital |
| 8. | P8 | 24 | Dalit | Grade 5 | Pyuthan-6 | 1 | Normal, Hospital |
| 9. | P9 | 27 | Brahmin | 10+2 | Pyuthan-4 | 3 | Normal, HP |
| Key Informant Interview | | | | | | | |
| S.N. | Code | Age (years) | Position | Working Organization | | Years of serving | Strata |
| 1. | V1 | 37 | FCHV | Tusara HP | | 12 | Jhimruk |
| 2. | V2 | 43 | FCHV | Dharmawati HP | | 17 | Pyuthan |
| 3. | V3 | 40 | FCHV | Sapdada HP | | 9 | Pyuthan |
| 4. | V4 | 56 | FCHV | Okharkot HP | | 32 | Jhimruk |
| 5. | A1 | 36 | ANM | Torwang HP | | 4 | Jhimruk |
| 6. | A2 | 44 | Sr. ANM | Pyuthan Hospital | | 18 | Pyuthan |
| 7. | H1 | 41 | HC | Jhimruk RM | | 4 | Jhimruk |
| 8. | H2 | 49 | HC | Pyuthan M | | 2 | Pyuthan |
| 9. | K | 38 | PHN | Health Office, Pyuthan | | 2.5 | Pyuthan District |

HP-Health Post, M-Municipality, RM-Rural Municipality

## 2.3 Sample size

The sample size of the study was calculated using the Cochrane equation [16] with a single proportion, taking into account the proportion of women having PNC check-ups as per the protocol (p) = 16%, as observed from the Annual Health Report 2018/19 published by the Department of Health Services (DoHS), Nepal [17]. Other parameters considered were the level of significance (α) as 5%, the margin of error (e) as 5%, the non-response rate (r) as 20% with taking reference from one of the telephone-based interviews conducted in Nepal [18], and design effect (DE) as 1.5.

$$Sample\ size\ (n_o) = \frac{z^2 pq}{e^2} * DE = 310$$

The expected number of live births in the selected clusters (N) is 1221. Therefore, the minimum sample size with finite population correction and non-response adjustment was calculated using the following formula:

$$Minimum\ Sample\ size\ (n1) = \left[\frac{n_o}{\left[1 + \frac{(n_0 - 1)}{N}\right]}\right] * (1 + 0.2) = 298$$

The calculated minimum sample size was 298. However, fifty participants were approached from each of the six selected clusters for data collection, resulting in a final sample size (n) of 300. Out of 300 approached, 243 postnatal mothers gave consent to participate in the study.

For qualitative component of the study, nine IDIs and nine KIIs were done. IDIs with postnatal mothers stopped after thematic information saturation.

## 2.4 Sampling method and data collection techniques

The Local units of Pyuthan district were first divided into two strata: urban municipalities and rural municipalities. Pyuthan urban municipality from urban and Jhimruk Rural municipality from rural stratum were selected randomly. Birthing Centers, Basic Emergency Obstetric and Neonatal Care (BEONC) sites and Comprehensive Emergency Obstetric and Neonatal Care (CEONC) sites within the selected local units were listed. Three health facilities from each of the stratum were then selected using probability proportionate to size (PPS) method (Table 2).

A sampling frame of postnatal mothers was prepared from selected health facility records and corresponding FCHV registers. Mothers were contacted via phone calls, attempted thrice a day for at least one week before being considered non-responsive. Quantitative data was collected using a structured questionnaire, while separate semi-structured open-ended interview guidelines were employed for in-depth and key-informant interviews.

## 2.5 Operational definition

**Caste:** Dalit, Janajati, Madhesi, Muslim, Brahmin/Chhetri, Other [19].

**Education:** No Education-without formal education; Below SLC-Grade 1 to 10 but not passed school leaving certificate (SLC); SLC and Above-SCL passed and above. **Knowledge on PNC:** Good with score more than median score and Poor with less than or equal to median score.

**Overall accessibility:** Accessible: if a postnatal mother had a health facility within a 30-minute walking distance, access to a motorable road, and a health worker was present during her last visit to the health facility during postpartum period. Not accessible: if one or all of these conditions were not fulfilled.

**Table 2. Sampling method for quantitative study.**

| Unit | Unit Selected | Selection Method | Cluster (Eligible Health Facility) | Cluster Selected | Selection Method |
|---|---|---|---|---|---|
| District | Pyuthan | Purposive | | | |
| **Strata** | | | | | |
| **Urban Municipality** | | | | | |
| Pyuthan | Pyuthan Urban Municipality | Random | Dakhakwadi HP | | PPS* (Based on expected live births) |
| | | | Sapdada HP | Sapdada HP | |
| | | | Pyuthan Hospital | Pyuthan Hospital | |
| | | | Jumrikanda HP | | |
| Swargadwari | | | Majhkot HP | | |
| | | | Dharmawati HP | Dharmawati HP | |
| | | | Maranthana HP | | |
| **Rural Municipality** | | | | | |
| Aairawati | Jhimaruk Rural Municipality | Random | Bangemaroth HP | | PPS* (Based on expected live births) |
| Gaumukhi | | | Okharkot HP | Okharkot HP | |
| Jhimaruk | | | Badikokt HP | | |
| Mallarani | | | Torwang HP | Torwang HP | |
| Mandavi | | | Tusara HP | Tusara HP | |
| Naubahini | | | | | |
| Sarumarani | | | | | |

* PPS-Probability Proportionate to Size

**Overall satisfaction:** Satisfied: if a postnatal mother was satisfied with both the behavior of the health worker and the service provided by the health facility; Not satisfied: if she was not satisfied with one or both conditions.

**PNC service utilization as per protocol:** Refers to the three postnatal check-ups within the first seven days after childbirth, i.e. first check-up within 24 hours, second on third day and third on the seventh day.

**Religion:** Hindu, Buddhist, Muslim, Kirat, Christian, Other [4]

**Socio-economic status (SES):** Upper with score more than or equal to 16 and lower with score less than 16 in Kuppuswamy's SES scale.

**Woman autonomy:** Defined in terms of a postnatal mother's ability to make decisions about her own health care during the last postnatal period. High autonomy: if a postnatal mother alone made decisions about her health care; Medium: if she along with her partner or other relatives, made decisions about her health care; No/Low autonomy: if her partner or someone else made decisions about her health care.

### 2.6 Conceptual framework

As outlined in the conceptual framework (Fig 1), the outcome variable was PNC service utilization as per protocol. The independent variables included respondent's socio-demographic characteristics, and the mediating variables encompassed obstetric characteristics, knowledge on PNC, women autonomy, health service-related factors and effect of Covid-19 on PNC service utilization.

### 2.7 Data analysis

Data analysis was carried out using STATA (version 17.0), presenting descriptive statistics through frequency, percentage, mean, and median. Multivariate logistic regression was carried

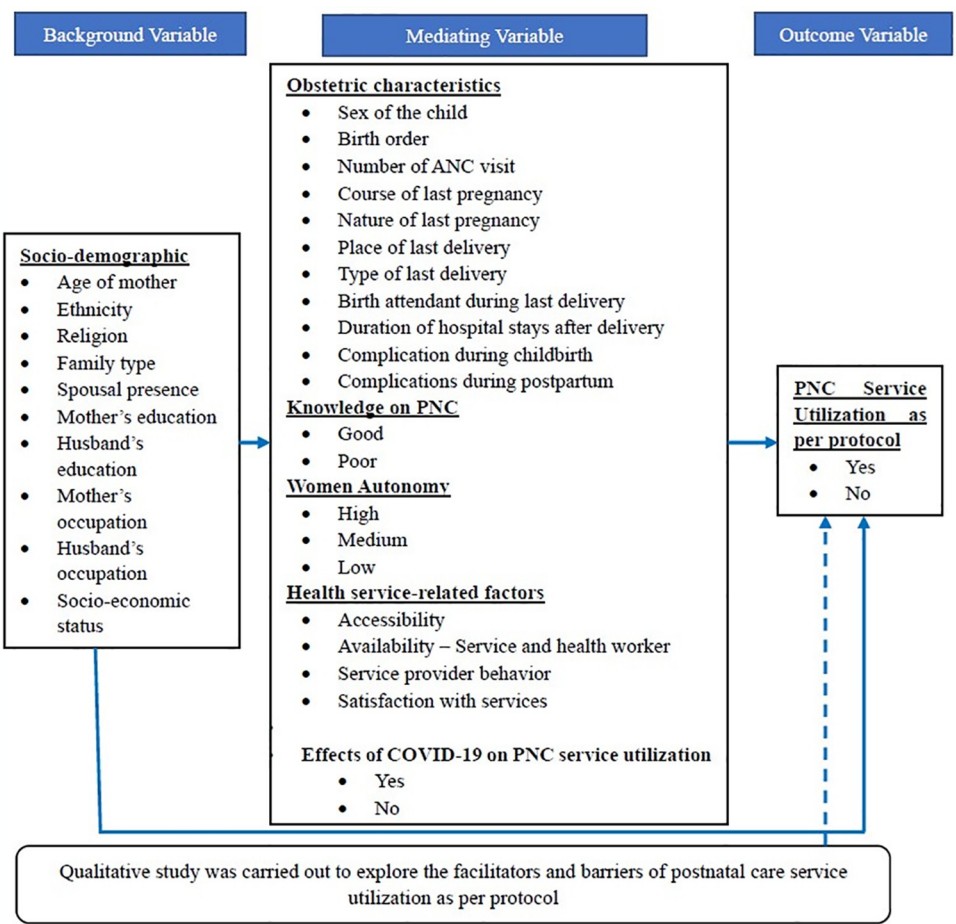

**Fig 1. Conceptual framework.**

out to determine outcome predictors. Multicollinearity was assessed using the variation inflation factor (VIF). Variables with VIF > 2 and p-value > 0.25 were excluded from the final model. Sampling design and weight were considered for unbiased estimates.

The IDIs and KIIs were audio recorded, transcribed, and then translated into English version. R-package for qualitative data analysis (RQDA) was used for qualitative data management. Braun and Clark's six-steps thematic analysis with inductive-deductive continuum was used for qualitative data analysis. The translated scripts were read and re-read to generate the codes. The codes were generated by two independent coders for the purpose of inter-coder agreement (ICA) analysis. A coding framework was developed based on two predetermined themes: facilitators and barriers. Codes for other themes, such as the nature of postpartum complications and knowledge of postpartum care, were identified inductively. The details of the coding and theme generation can be found in the supporting information file (S3 File. Coding framework). The findings from quantitative and qualitative analysis were triangulated to identify the area of convergence and divergence.

## 2.8 Validity and reliability

The questionnaire, developed through extensive literature review, underwent content validation via discussions with the guide, co-guide, and faculty at the School of Public Health of

Patan Academy of Health Sciences (PAHS). Clinical contents were validated by a Gynecologist. After pre-testing with 15 randomly selected mothers at Pyuthan District Hospital, necessary modifications were made. Forward and backward translation was done to ensure the reliability of the tool.

Iterative questioning was done during IDIs and KIIs to generate in-depth information. Member checking was done with selected participants (P4, V1 and H1) to ensure the credibility of information. An independent intercoder positive percent agreement was calculated for the reliability of the codes and was found as 77.55%.

## 2.9 Ethical consideration

Ethical approval for this study was obtained from the Institutional Review Committee–Patan Academy of Health Sciences (Ref: PHP2107201557). Additionally, approvals were obtained from Pyuthan Municipality (Ref. No.- 2384), Jhimruk Rural Municipality (Ref. No.-1598) before data collection. All participants were briefed on the study's objectives, and their voluntary participation was ensured. They were encouraged to ask questions about the study and were informed that they had the right to withdraw from the interview at any time without providing any explanation. Participants were explicitly informed that there were no direct benefits or harms associated with participating in this study. Verbal informed consent was obtained before starting the interview. The entire process, from explaining the study's objectives to obtaining verbal consent, was audio recorded for each participant. A unique ID was assigned while keeping the record of each verbal consent. The telephonic method of interview, along with obtaining verbal consent, was approved by the Institutional Review Committee of Patan Academy of Health Sciences. Since all the participants were 18 years old or older, an assent form was not used during interviews.

# 3. Results

## 3.1 Socio-demographic characteristics

The study had an 80% overall response rate. Participants had a median age of 23 years, with an interquartile range of 5. Majority of the participants were Brahmin/Chhetri (49.88%), followed the Hindu religion (98.75%), lived in joint families (64.95%) and had their husbands present with them (67.80%) during their last postpartum period (Table 3).

## 3.2 Obstetric characteristics

Over half the participants (51.48%) were first-time mothers, with 56.28% giving birth to boys. Over 95.45% of pregnancies were wanted. Almost all deliveries (97.11%) occurred in health facilities, assisted by health workers (97.12%). A quarter (25.06%) experienced complications during delivery, and about one fifth (18.91%) faced postpartum complications (Table 4).

## 3.3 Complications during pregnancy, delivery and postpartum

The highest reported complication during pregnancy was lower abdominal pain (35.42%), followed by swelling of face or hand (22.92%). Prolonged labor (64.38%) was the most reported complication followed by severe vaginal bleeding (6.85%) during delivery. Most reported postpartum complications were severe bleeding (52.73%), pre-eclampsia or eclampsia (7.27%) and wound infection (7.27%) (Table 5).

During qualitative interviews, participants highlighted that two of the recent maternal deaths occurred due to postpartum complication.

**Table 3. Sociodemographic characteristics of the respondents (unweighted and weighted analysis).**

| Variables | Unweighted (n = 243) | | Weighted (n = 243) | |
|---|---|---|---|---|
| | Frequency | Percentage | Frequency | Percentage |
| Age (Median and IQR) | 23 [IQR–5] | | Min–18, Max–44 | |
| **Ethnicity** | | | | |
| Dalit | 60 | 24.69 | 59.80 | 24.61 |
| Janajati | 61 | 25.10 | 61.04 | 25.11 |
| Madhesi | 1 | 0.41 | 0.95 | 0.39 |
| Brahmin/Chhetri | 121 | 49.79 | 121.20 | 49.88 |
| **Religion** | | | | |
| Hindu | 240 | 98.77 | 239.97 | 98.75 |
| Other* | 3 | 1.23 | 3.03 | 1.25 |
| **Family Type** | | | | |
| Nuclear | 85 | 34.98 | 85.18 | 35.05 |
| Joint | 158 | 65.02 | 157.82 | 64.95 |
| **Presence of husband during postpartum period** | | | | |
| Yes | 165 | 67.90 | 164.76 | 67.80 |
| No | 78 | 32.10 | 78.24 | 32.20 |
| **Education of mother** | | | | |
| No Education | 22 | 9.05 | 22.08 | 9.09 |
| Below SLC | 126 | 51.85 | 126.05 | 51.87 |
| SLC and above | 95 | 39.09 | 94.87 | 39.04 |
| **Education of husband** | | | | |
| No Education | 7 | 2.88 | 6.96 | 2.86 |
| Below SLC | 130 | 53.50 | 130.18 | 53.57 |
| SLC and above | 106 | 43.62 | 105.86 | 43.57 |
| **Occupation of mother** | | | | |
| Employed | 53 | 21.81 | 52.57 | 21.63 |
| Unemployed | 190 | 78.19 | 190.43 | 78.37 |
| **Occupation of husband** | | | | |
| Employed | 228 | 93.83 | 227.81 | 93.75 |
| Unemployed | 15 | 6.17 | 15.19 | 6.25 |
| **Socioeconomic status** | | | | |
| Upper | 40 | 16.46 | 40.17 | 16.53 |
| Lower | 203 | 83.54 | 202.83 | 83.47 |

"*The maternal deaths occurred within the district were both during postpartum period. One died on the way to district hospital after referral from health post due to uterus inversion following delivery. Another was due to postpartum hemorrhage. One pregnant woman died of COVID-19 this year.*"

'*K*'

## 3.4 Autonomy and PNC knowledge

Knowledge on PNC was assessed based on knowledge about postnatal period, PNC check-up, PNC services and postnatal danger signs. The median knowledge score was 3 (IQR -2), and approximately 40% (39.66%) of postnatal mothers were found to have good PNC knowledge.

**Table 4. Obstetric characteristics of the respondents (n = 243, weighted).**

| Variables | Frequency | Percentage |
|---|---|---|
| **Sex of the child** | | |
| Male | 137 | 56.28 |
| Female | 106 | 43.72 |
| **Birth order** | | |
| First | 125 | 51.48 |
| Second or third | 97 | 39.87 |
| Fourth or more | 21 | 8.66 |
| **Number of ANC (Median and IQR)** | 4 (IQR– 2) | |
| **Course of Last Pregnancy** | | |
| With Complications | 39 | 16.09 |
| Without Complications | 204 | 83.91 |
| **Nature of last pregnancy** | | |
| Wanted | 232 | 95.45 |
| Unwanted | 11 | 4.55 |
| **Place of Last Delivery** | | |
| Home Delivery | 3 | 1.24 |
| Birthing Center | 123 | 50.76 |
| Comprehensive emergency obstetric and neonatal care (CEONC) | 113 | 46.35 |
| Other | 4 | 1.65 |
| **Type of Delivery** | | |
| Spontaneous Vaginal Delivery | 219 | 90.17 |
| Assisted Vaginal Delivery | 4 | 1.64 |
| Cesarian Section | 20 | 8.19 |
| **Birth Attendant during last delivery** | | |
| Doctor | 26 | 10.66 |
| Nurse | 81 | 33.15 |
| ANM | 129 | 53.31 |
| Friends/Relatives/Neighbor | 7 | 2.88 |
| **Hospital Stay After Delivery** | | |
| ≤ 6 hours | 62 | 25.63 |
| 7–24 hours | 105 | 43.32 |
| > 24 hours | 73 | 29.82 |
| Not Applicable | 3 | 1.24 |
| **Presence of complications during delivery** | | |
| Yes | 63 | 26.05 |
| No | 180 | 73.95 |
| **Complications during postpartum period** | | |
| Yes | 46 | 18.91 |
| No | 197 | 81.09 |

Healthcare decisions were made independently by only around one-fifth (19.50%) of postnatal mothers during their last postnatal period (Table 6).

During qualitative interviews only a few postnatal mothers were able to correctly define the PNC check-up as per protocol and postnatal period. The determination of the postnatal period in rural areas was based on the postnatal mother's ability to return to normal work rather than potential health risks during the postpartum period.

**Table 5. Complications during pregnancy, delivery and postpartum.**

| Types of Complications (Respondents reported) | Frequency of responses | Percentage of responses | Percentage of cases |
|---|---|---|---|
| *Pregnancy (Total response = 48, n = 39, multiple response, non-weighted)* | | | |
| Lower abdominal pain | 17 | 35.42 | 43.59 |
| Swelling in face or hand | 11 | 22.92 | 28.21 |
| Vaginal bleeding | 7 | 14.58 | 17.95 |
| Severe headache | 2 | 4.17 | 5.13 |
| Pre-eclampsia/eclampsia | 1 | 2.08 | 2.56 |
| High blood pressure | 1 | 2.08 | 2.56 |
| Other | 9 | 18.75 | 23.08 |
| *Delivery (Total response = 73, n = 63, multiple response, non-weighted)* | | | |
| Prolonged labor | 47 | 64.38 | 73.44 |
| Severe vaginal bleeding | 5 | 6.85 | 7.81 |
| Pre-term labor | 4 | 5.48 | 6.25 |
| Abnormal position of the fetus | 3 | 4.11 | 4.69 |
| Retained placenta | 2 | 2.74 | 3.13 |
| Fever | 1 | 1.37 | 1.56 |
| Other | 11 | 15.07 | 17.19 |
| *Postpartum (Total response = 55, n = 46, multiple response, non-weighted)* | | | |
| Postpartum severe bleeding | 29 | 52.73 | 63.04 |
| Pre-eclampsia or eclampsia | 4 | 7.27 | 8.70 |
| Infection of wound | 4 | 7.27 | 8.70 |
| Septicemia | 2 | 3.64 | 4.35 |
| Increased pain or infection in the perinium | 2 | 3.64 | 4.35 |
| High fever | 1 | 1.82 | 2.17 |
| Severe headache | 1 | 1.82 | 2.17 |
| Depression or suicidal thought | 1 | 1.82 | 2.17 |
| Other | 11 | 20.0 | 23.91 |

"*I think there are three check-ups: the first takes place in the hospital within 24 hours, the second on the third day, and the third on the seventh day.*"

'*P5*'

*We usually consider postpartum period for about 3 months. After that, we start getting back to our regular household chores. However, we don't rush into returning to work leaving the baby at home. It takes a bit longer to get back into the job.*"

'*P9*'

**Table 6. Autonomy and knowledge on PNC (n = 243, weighted).**

| Variables | Frequency | Percentage |
|---|---|---|
| **Median knowledge score and interquartile range** | **3 (IQR—2)** | |
| **Knowledge on PNC** | | |
| Good | 96 | 39.66 |
| Poor | 147 | 60.34 |
| **Autonomy in health care decision making** | | |
| High | 47 | 19.50 |
| Medium | 77 | 31.53 |
| Low | 119 | 48.97 |

**Table 7. Postnatal mothers' access to and satisfaction with health services (n = 243, weighted).**

| Variables | Frequency | Percentage |
|---|---|---|
| **Health service accessibility** | | |
| *Walking distance of nearest health facility in minute (average)* | **55.15 minutes (SD– 41.63), Min– 2, Max– 180** | |
| *Access of motorable road* | | |
| Yes | 204 | 84.04 |
| No | 39 | 15.96 |
| *Presence of health worker* | | |
| Yes | 234 | 96.32 |
| No | 9 | 3.68 |
| *Overall accessibility* | | |
| Yes | 107 | 43.99 |
| No | 136 | 56.01 |
| **Satisfaction with health services** | | |
| *Satisfaction with health worker behavior* | | |
| Yes | 226 | 93.03 |
| No | 17 | 6.97 |
| *Satisfaction with service provided* | | |
| Yes | 202 | 83.19 |
| No | 41 | 16.81 |
| *Overall Satisfaction* | | |
| Yes | 201 | 82.76 |
| No | 42 | 17.24 |
| **Affected to receive PNC services due to COVID-19** | | |
| Yes | 42 | 17.20 |
| No | 201 | 82.80 |

## 3.5 Health service-related factors

The average walking distance to health facilities was 55.15 minutes (SD-41.63 minutes). Overall accessibility, which was considered accessible if a postnatal mother had a health facility within a 30-minute walking distance, access to motorable road, and a health worker was present during her last visit to the health facility during postpartum period, was 43.99%. The overall satisfaction, which was considered satisfied if a postnatal mother was satisfied with both the behavior of health worker and ther services provided by health facility, was 82.76%. More than four-fifths (82.80%) reported that the COVID-19 pandemic did not affect them to receive postnatal care services (Table 7).

## 3.6 PNC service utilization

The overall prevalence of PNC service utilization as per protocol in the district was 38.43% (95% CI: 32.48–44.74). Out of 123 postnatal mothers participated from Pyuthan Municipality, 32.52% (95% CI: 20.67–47.66) had PNC check-up as per protocol, while 44.17% (95% CI: 27.57–62.32) among 120 postnatal mothers from Jhimruk Rural Municipality had PNC check-up as per protocol (Table 8).

**Table 8. Prevalence of PNC service utilization (n = 243, weighted).**

| Variables | Frequency | Percentage [95% CI] |
|---|---|---|
| **PNC check-up as per protocol** | | |
| Pyuthan District | 93 | 38.43 [32.48–44.74] |
| Pyuthan Municipality | 40 | 32.52 [20.67–47.66] |
| Jhimruk Rural Municipality | 53 | 44.17 [27.57–62.32] |

## 3.7 Factors affecting PNC service utilization

**3.7.1 Bivariate analysis of outcome variables with background and mediating variables.** In bivariate analysis, postnatal mothers with education SLC and above, those residing in families with higher socioeconomic status, having four or more ANC check-ups, possessing good knowledge on PNC, having access to motorable roads, and not being affected to receive service due to COVID-19 pandemic were associated with a higher likelihood of utilizing PNC services as per protocol. Only variables found significant ($p<0.05$) during bivariate analysis were presented in the table (Table 9).

**Table 9. Bivariate analysis with background (model-I) and mediating variables (model-II) (Weighted).**

| Variables | Unadjusted OR [95% CI] | p-value |
|---|---|---|
| **Background Variables** | | |
| **Education level of Postnatal Mother** | | **0.01\*** |
| No Education | 1.0 | |
| Below SLC | 2.64 [0.84–8.32] | |
| SLC and Above | **3.78 [1.56–9.18]** | |
| **Socio-economic status** | | **0.00\*\*** |
| Lower | 1.0 | |
| Upper | **4.30 [3.61–5.12]** | |
| **Mediating variables** | | |
| **Number of ANC check-up during last pregnancy** | | **0.04\*** |
| Less than four | 1.0 | |
| Four or more | **1.5 [1.01–2.23]** | |
| **Knowledge on PNC** | | **0.002\*\*** |
| Poor | 1.0 | |
| Good | **7.35 [3.34–15.72]** | |
| **Women's autonomy** | | **0.015\*** |
| Low | 1.0 | |
| Medium | 2.12 [1.38–3.24] | |
| High | 1.71 [1.09–2.66] | |
| **Access of motorable road** | | **0.001\*** |
| No | 1.0 | |
| Yes | **6.81 [3.59–12.92]** | |
| **Affected to receive services due to COVID-19** | | **0.03\*** |
| Yes | 1.0 | |
| No | **3.14 [1.13–8.66]** | |

\* $<0.05$,

\*\* $<0.01$

**3.7.2 Multivariate logistic regression of eligible background and mediating variables.**
Multivariate logistic regression of each of the eligible background variables and mediating variables was conducted separately to observe their effect on outcome variable. The variables with VIF < 2 and p-value < 0.25 were considered as eligible variables and qualified for multivariate regression. After checking for multicollinearity, education level of husband and socio-economic status of the family were found eligible from background variables while sex of current born child, nature of last pregnancy, place of delivery, type of delivery, knowledge on PNC, access of motorable road, overall satisfaction, and affected to receive services due to COVID-19 pandemic were found eligible from mediating variables. The multivariate logistic regression revealed that postnatal mothers resided in higher socioeconomic families (AOR-3.19; CI: 2.14–4.77), delivered in birthing centers (AOR-2.75; CI: 2.04–3.72), had good PNC knowledge (AOR-7.46; CI: 4.44–12.57) and had access to motorable roads (AOR-8.04; CI: 6.29–10.26) were associated with a higher likelihood of utilizing PNC services as per protocol (Table 10).

**3.7.3 Predictors of PNC service utilization as per protocol.** Final multivariate logistic regression of significant background and mediating variables from Model-III and Model-IV with outcome variable was performed to identify the independent predictors of PNC service utilization as per protocol. The multicollinearity test was performed before the final model was run, and no multicollinearity issue was observed between significant variables from Model-III and Model-IV. The final model revealed that postnatal mothers from families with higher socioeconomic status were 3.84 times (AOR-3.84; CI: 2.40–6.15) more likely to use PNC services than those in lower socioeconomic families, controlling for all other variables. Similarly, women delivering in birthing centers were 1.86 times (AOR-1.86; CI: 1.16–3.00) more likely to utilize services compared to those delivering at home or en route to a health facility. Likewise, Mothers with good PNC knowledge were 6.75 times (AOR-6.75; CI: 3.39–13.45) and those with access to motorable roads were 6.30 times (AOR-6.30; CI: 3.94–10.08) more likely to use PNC services as per protocol than those with poor knowledge and no access to motorable roads, respectively, after adjusting for all other variables (Table 11).

The goodness-of-fit test with p-value more than 0.05 (P—0.14) indicates that the model fits the data well, and the Area Under the Curve (AUC) value of 81.94% (95% CI: 76.39–87.49) indicates a reliable model.

## 3.8 Facilitators and barriers of PNC service utilization–Findings from qualitative study

**3.8.1 Facilitators.** *a. PNC home visit.* The PNC home visit program was implemented in the Pyuthan district last fiscal year and was deemed crucial for improving PNC check-ups as per protocol.

> *"There has been a significant shift in service utilization due to PNC home visit program. Prior to this, only mothers who delivered in a health facility received their first PNC check-up before being discharged. There was no follow-up check-up on the third and seventh days."*

'V1'

> *"For the second and third check-ups, they came to my home on the third and seventh day, respectively."*

'P4'

Before the implementation of the PNC Home Visit Program, health workers could not follow up on the second and third days for postnatal check-ups of mothers due to financial

**Table 10. Multivariate logistic regression of eligible background variables- Model III and mediating variables—Model IV (weighted).**

| Variables | Unadjusted OR [95% CI] | P-value | Adjusted OR [95% CI] | P-value |
|---|---|---|---|---|
| *Background Variables* | | | | |
| **Education level of Husband** | | | | |
| No Education | 1.0 | | 1.0 | |
| Below SLC | 1.07 [0.14–8.26] | 0.93 | 1.06 [0.14–8.15] | 0.94 |
| SLC and above | 2.58 [0.34–19.67] | 0.26 | 1.69 [0.22–12.79] | 0.51 |
| **Socio-economic status** | | | | |
| Lower | 1.0 | | 1.0 | |
| Upper | **4.30 [3.61–5.12]** | **0.00**\*\* | **3.19 [2.14–4.77]** | **0.001**\*\* |
| *Mediating Variables* | | | | |
| **Sex of current born child** | | | | |
| Female | 1.0 | | 1.0 | |
| Male | 1.12 [0.80–1.56] | 0.41 | 1.51 [0.95–2.38] | 0.07 |
| **Nature of last pregnancy** | | | | |
| Unwanted | 1.0 | | 1.0 | |
| Wanted | 6.83 [0.26–179.02] | 0.17 | 5.08 [0.11–228.3] | 0.30 |
| **Place of delivery** | | | | |
| Non-institutional | 1.0 | | 1.0 | |
| Birthing center | 2.15 [0.76–6.10] | 0.11 | **2.75 [2.04–3.72]** | **0.001**\*\* |
| CEONC | 1.05 [0.25–4.43] | 0.93 | 1.35 [0.64–2.84] | 0.32 |
| **Type of delivery** | | | | |
| Normal Vaginal Delivery | 1.0 | | 1.0 | |
| CS/Assisted Delivery | 0.49 [0.08–2.92] | 0.33 | 0.30 [0.04–2.35] | 0.18 |
| **Knowledge on PNC** | | | | |
| Poor | 1.0 | | 1.0 | |
| Good | **7.35 [3.34–15.72]** | **0.002**\*\* | **7.46 [4.44–12.57]** | **0.000**\*\* |
| **Access of motorable road** | | | | |
| No | 1.0 | | 1.0 | |
| Yes | **6.81 [3.59–12.92]** | **0.001**\*\* | **8.04 [6.29–10.26]** | **0.000**\*\* |
| **Overall satisfaction** | | | | |
| No | 1.0 | | 1.0 | |
| Yes | 1.94 [0.43–8.74] | 0.29 | 2.47 [0.38–16.18] | 0.25 |
| **Affected to receive services due to COVID-19** | | | | |
| Yes | 1.0 | | 1.0 | |
| No | 3.14 [1.13–8.66] | **0.03**\* | 2.94 [0.57–15.07] | 0.14 |

\* <0.05,

\*\* <0.01

constraints. After the implementation of the PNC Home Visit Program, three PNC check-ups as per protocol were made mandatory in the district, and they started receiving conditional grants from the federal government to run the program. Additionally, local governments also began taking initiatives to improve PNC service utilization at the local level.

*"Previously, it was difficult. We were not able to increase it from 17%. Postnatal mothers did not want to visit health facilities for check-ups, and we didn't have a provision to incentivize health workers for home visits. Later, we discussed the issue with local authorities and health coordinators, and they decided to set aside funds to provide incentives to health professionals*

**Table 11. Multivariate logistic regression of significant background and mediating variables-Final Model (weighted).**

| Variables | Adjusted OR [95% CI] | P-value |
|---|---|---|
| **Socio-economic status** | | |
| Lower | 1.0 | |
| Upper | **3.84 [2.40–6.15]** | **0.001**\*\* |
| **Place of delivery** | | |
| Non-institutional | 1.0 | |
| Birthing center | **1.86 [1.16–3.00]** | **0.02**\* |
| CEONC | 0.64 [0.24–1.71] | 0.27 |
| **Knowledge on PNC** | | |
| Poor | 1.0 | |
| Good | **6.75 [3.39–13.45]** | **0.002**\*\* |
| **Access of motorable road** | | |
| No | 1.0 | |
| Yes | **6.30 [3.94–10.08]** | **0.000**\*\* |

\* <0.05,

\*\* <0.01

*in the form of reimbursements for travel expenses for home visits. Local governments allocated funds for this in the last fiscal year, and we also received conditional grant from the federal government's PNC Home Visit Program."*

'K'

*b. Incentives.* Incentives, such as transportation costs under the Aama Suraksha program provided by the federal government along with additional incentives from local government bodies, facilitated the improvement of PNC service utilization.

*"Even though it's not much, the money provided by the Government of Nepal for mothers who delivered at health facility may be of little help."*

'V1'

*"Our municipality has given postpartum mothers a nutrition and hygiene package, which includes clothes, Horlicks, sanitary pads, eggs, and other items. This has helped them to seek health services during the postpartum period."*

'H2'

*"Rather than the health office, local governments have different programs to improve PNC services. Some local governments provide mattresses to postnatal mothers, while others provide baby kits. Some local authorities have specific programs like "Municipality with Postnatal Mothers." These types of programs have really helped to improve institutional delivery and thereby PNC check-ups as per protocol. In cases of home delivery, FCHVs immediately inform health workers, who then visit the postnatal mother's home for check-ups."*

'K'

*c. Counselling by health workers.* Counseling by health workers was found helpful in raising awareness about postpartum care and services among postnatal mothers and their families, thereby increasing service seeking and utilization.

"*Yes, I knew about the services provided at the health facility for postnatal mothers. I'm part of a health mothers' group where FCHV informed us about these and that helped me to go for PNC check-up during my postnatal period.*"

'*P9*'

"*The doctors at the district hospital advised me to breastfeed my child soon after birth and to go for the second and third check-ups at the nearest health facility.*"

'*P3*'

*d. Awareness and family support.* Awareness among postnatal mothers and their family members was identified as a significant facilitator for PNC service utilization. Participants during interviews pointed out that those who were aware were more likely to seek services. The support from family members was equally important.

"*Postnatal check-ups are done by those who believe the check-up is for themselves and their newborn babies.*"

'*P9*'

"*Families are now more aware of the importance of caring for mothers and babies, and they understand that keeping the mother and baby healthy is beneficial to the entire family.*"

'*V1*'

*e. Proximity of health facility.* Proximity of health facilities was reported to be vital in utilization of postnatal care health services during qualitative interviews.

"*It's about 10 minutes far. We went because we lived near to health facility.*"

'*P4*'

"*The health post is near to Gejbang and Khaprengkhola, and mothers living in these areas don't need transportation to reach health post. However, the health post is far from Rajyan, Managau, and Bhendabari, and there is no motorable road for ambulance either.*"

'*V2*'

**3.8.1 Barriers.** *a. Geographical and transportation difficulty.* Geographical and transportation difficulties were reported as major barriers during interviews. Postnatal mothers were found not seeking health services due to the far distance of the health facility. Furthermore, health workers also expressed challenges in reaching the homes of postnatal mothers to provide services.

"*It wasn't possible to go for a PNC check-up. We had no choice but to go for the delivery; otherwise, we would have refused to go. It's too far from where we live. That's why, after giving birth, I didn't go for a single check-up.*"

'P3'

"*The distance to the health facility, hilly and sloped areas, etc., are the main barriers preventing postnatal mothers from accessing health services. During our PNC visit, it takes roughly five to six hours, or a full day, to reach one postnatal mother.*"

'A1'

b. *Postpartum weakness to visit health facility*. Postpartum weakness was another barrier for seeking services during the postpartum period, as mothers felt too weak to visit the health facility in the early days of postpartum.

"*If I needed to go to the health post for a check-up on the third day of my postpartum period, I wouldn't have been able to because I couldn't get there on foot.*"

'P4'

c. *Lack of awareness*. Some postnatal mothers mentioned in interviews that they were completely unaware of the PNC check-up. Another barrier identified was the perceived low importance of care during postpartum period.

"*First and foremost, I wasn't aware that we should also do PNC check-ups after giving birth to the baby.*"

'P6'

"*People believe that women can have issues throughout pregnancy and delivery only, and they have no complications after giving birth.*"

'K'

c. *COVID-19 pandemic*. Fear of COVID-19 transmission, either from health facility or through vehicles used for transportation, were major issues reported during interviews, making postnatal mothers difficult to visit health facilities.

"*COVID had a significant impact; it was difficult to go outside. Some mothers missed their check-ups, and some did not go for check-ups because they were afraid of COVID transmission.*"

'P4'

"*They could use the district hospital's ambulances, but there was a risk of transmission because ambulances were also used by COVID-19 patients.*"

'V1'

d. *Difficult convincing elderly family members*. During interviews, health workers revealed difficulty in convincing senior citizens in postnatal mothers' homes to take care of postnatal mothers.

"*It is difficult to convince older people. They argue that there was nothing unusual about this period to name it special because they had already gone through this phase.*"

'V2'

*e. Financial challenges*. Financial difficulty during delivery and postpartum period was reported by postnatal mothers during interviews. In some cases, loans were taken from different people and organizations to manage expenses.

"*During that time, we had a lot of expenses. I borrowed money from different people and organizations. I did whatever I could to manage expenses.*"

'*P5*'

"*I borrowed money from various people and cooperative organizations; no one donates for free, sir.*"

'*P2*'

*f. Inadequate health workforce*. During interviews, health personnel highlighted that the current workforce is inadequate to run all the programs at the community level. Health workers were unable to run the PNC home visit program as intended due to a lack of adequate health workforce.

"*Health personnel cannot perform PNC home visits as per protocol because there are few health workers to handle a high number of cases, and patients may arrive for delivery at any time. As a result, PNC check-ups are not being carried out according to protocol.*"

'*V1*'

"*We have to run the expanded program with the same number of employees, and it's challenging. In addition to our regular duties, we must attend PNC visits, senior citizen visits, outreach clinics, EPI clinics, monitoring, and supervision, and the COVID campaign, among other things.*"

'*H1*'

## 3.9 Triangulation of the quantitative and qualitative findings

The findings from the quantitative and qualitative studies were triangulated to identify areas of convergence and divergence. The quantitative findings from multivariate logistic regression and individual responses on facilitators and barriers during the quantitative survey, along with qualitative sub-themes of facilitators and barriers, are presented in Table 12.

**3.9.1 Areas of convergence.** Good knowledge of PNC was identified as an independent predictor of PNC service utilization in quantitative analysis whereas lack of awareness was identified as a barrier. The combined quantitative and qualitative findings affirm the significance of knowledge or awareness of postnatal care in increasing PNC service utilization.

Access to a motorable road was identified as an independent predictor of PNC service utilization in multivariate logistic regression whereas in qualitative analysis, the lack of transportation facilities, encompassing both roads and vehicles, was recognized as a barrier to PNC service utilization.

Home-based services through PNC home visits were the most reported facilitator of PNC service utilization during quantitative survey. Qualitative analysis also identified PNC home visits as a prominent factor for enhanced PNC service utilization in the district.

Postpartum weakness to return to the health facility for check-ups was the most reported barrier during quantitative survey, with qualitative analysis supporting finding. Postnatal

**Table 12. Triangulation of quantitative and qualitative findings.**

| Quantitative findings | | | Qualitative Findings |
|---|---|---|---|
| *Facilitators* * *(Total response = 582)* | *Percentage of Responses* | | *Facilitators* |
| *PNC services at doorstep* | 35.61 | | *PNC home visit* |
| *Awareness about PNC check-up* | 27.95 | | *Awareness and family support* |
| *Health facility nearby house* | 13.64 | | *Proximity of health facility* |
| *Getting informed by health worker* | 11.15 | | *Counselling by health worker* |
| *New-born illness* | 4.33 | | *Incentives* |
| *Family support* | 4.16 | | |
| *Self-illness* | 2.0 | | |
| *Other* | 1.16 | | |
| *Barriers* * *(Total response = 601)* | | | *Barriers* |
| *Felt weak to visit health facility* | 30.93 | | *Postpartum weakness to visit health facility* |
| *Not aware about PNC check-up* | 29.21 | | *Lack of awareness* |
| *Lack of transportation facility* | 25.60 | | *Geographical and transportation difficulty* |
| *Felt not necessary* | 7.22 | | *Financial challenges* |
| *No one to take care of newborn* | 3.09 | | *Difficult convincing elderly family members* |
| *Other* | 3.95 | | *Inadequate health workforce* |
| **Findings from multivariate logistic regression** | | | |
| *Socio-economic status (AOR: 3.84, CI: 2.40–6.15)* | | | |
| *Knowledge on PNC (AOR: 6.75, CI: 3.39–13.45)* | | | |
| *Access of motorable road (AOR: 6.30, CI: 3.94–10.08)* | | | |
| *Place of delivery-Birthing center (AOR: 1.86, CI: 1.16–3.00)* | | | |

\* Multiple response-unweighted

mothers disclosed that they felt too weak to go back to the health facility for check-ups during the early days of the postpartum period.

**3.9.2 Areas of divergence.** Participants in qualitative interviews mentioned the proximity of health facilities as a facilitator for the use of PNC services. However, during quantitative analysis, no significant association was observed between the proximity (distance) of health facilities and PNC check-ups as per protocol.

The COVID-19 pandemic was identified as the major barrier during qualitative analysis. However, it was not recognized as an independent predictor of PNC service utilization as per protocol in the multivariate logistic regression.

# 4. Discussion

## 4.1 Prevalence of PNC service utilization

The study showed that 38.43% of postnatal mothers have utilized PNC services as recommended by the government of Nepal. When compared to statistics from the fiscal year 2019/20 ((18.8%), the prevalence is more than twice as high as the national average [20]. In contrast to the findings from other studies conducted in Nepal, [13, 14, 21] the prevalence of PNC check-up as per protocol is relatively high in our study. This might be because the other studies were conducted before the implementation of the PNC home-visit program in Nepal. However, the increment was only 40.76% in contrast to the 68% overall increment in PNC check-ups as per protocol in PNC home visit-implemented municipalities [22]. Moreover, the findings of our study are somewhat similar to the results of a systematic review regarding factors affecting

postnatal care utilization in developing countries, in which the weighted prevalence of postnatal service utilization was 36% [23].

## 4.2 Factors affecting PNC service utilization

Our study found no significant association between participants' socio-demographic characteristics and PNC service utilization. In contrast, a study conducted in Baglung municipality identified significant associations of ethnicity and husband's occupation with complete PNC check-ups [13]. Another study in Ramechhap district revealed significant associations with mother's age, husband's education, husband's occupation, and family type in postnatal service utilization [24]. The universal nature of the safe motherhood program and local government initiatives may have facilitated postnatal mothers' access to health services regardless of their socio-demographic characteristics. Consistent with our findings, the Nepal Demographic and Health Survey 2011 indicated that postnatal mothers from rich families were more likely to use PNC services [25]. However, our results differ from studies conducted in India, [26–28] Indonesia, [29, 30] China, [31] Ethiopia [32, 33] and Nigeria, [34] which demonstrated associations between participants' socio-demographic characteristics and PNC service utilization. These discrepancies may be attributed to geographical variations and differences in participants' background characteristics, such as occupation and education of postnatal mothers, as observed in those studies.

The counseling for PNC check-ups during ANC visits, found to be associated with PNC check-ups in studies conducted in Nepal, [25] India, [26] Indonesia, [29] Nigeria, [35] and Tanzania [36] was reported to be low in our study. In the absence of effective counseling, the number of ANC visits may not have translated into postnatal check-ups.

## 4.3 Knowledge, autonomy, and health service-related factors

Our study identified knowledge of PNC as an independent predictor of PNC service utilization, contrasting with a study conducted in Baglung, Nepal, [13] where no association was observed. However, our results align with studies conducted in Ethiopia, [37–39] Malawi, [40] and Nigeria, [34] all indicating an association between postnatal mothers' knowledge and the utilization of PNC services. This highlights the importance of postnatal care knowledge in motivating mothers to prioritize their health and seek services promptly.

Maternal autonomy in postpartum health decisions showed no association with PNC check-ups in our study, consistent with one of the studies conducted in Nepal [25]. Nonetheless, the importance of women's autonomy is evident, with studies in Malawi and Ethiopia showing a positive association between women's decision-making autonomy and PNC service utilization.

Around 54% of the people in our study did not have access to the nearest health facility within walking distance of 30 minutes which is 23% less than the national average (77.1%) but when compared to the hilly region (66.3%), the difference was only 12% [4]. Though distance to health facilities was not identified as the predictors of PNC service utilization, access to motorable road was significantly contributing for service utilization in our study. The finding is consistent with the findings of Nepal health facility survey, which stated that when a road network and transportation facilities are available, health service utilization is better in the hilly and mountain regions than in the Terai region [41]. Similar kinds of findings was observed in studies conducted in Yemen [42] and North Dakota [43]. This preference for vehicular transport to health facilities, rather than walking, may contribute for seeking PNC services. The COVID-19 pandemic did not significantly affect PNC service utilization in our study, consistent with research in Nepal [44] and Ethiopia [45]. Although a decrease in PNC

services was reported, it did not reach statistical significance. This may be attributed to health staff providing services to some extent from health facilities or through home visits, as revealed in qualitative interviews.

## 4.4 Facilitators and barriers of PNC service utilization

The PNC Home Visit program identified as a crucial facilitator in enhancing the utilization of postnatal care services according to protocol in our study. Our finding was consistent with a systematic review regarding barriers and facilitators to postnatal care service utilization in low- and middle-income countries. The review stressed the role of government initiatives, emphasizing how PNC focused programs can effectively enhance the utilization of postnatal care services [46]. The findings were also supported by the study conducted in Rwanda where postnatal home visits by community health workers were well accepted and valued [47]. Consistent with our findings, family support and preparedness were also identified as facilitators for the utilization of maternity services, including antenatal and postnatal services, in a narrative review conducted in Nepal [48]. The incentives and packages offered for postnatal mothers were identified as factors contributing to the improvement of postnatal care (PNC) checkups in our study. This finding aligns with a study conducted in the Chitwan district of Nepal, which revealed that financial and material incentives provided to mothers encouraged them to utilize the services [49]. A study conducted in Indonesia supports our findings, indicating that conditional cash transfers for mothers were an effective alternative strategy for improving the utilization of maternal services in Indonesia [50]. The presence of a nearby health facility was identified as one of the major factors contributing to the improved utilization of maternal services in Nepal [49], which aligns with the findings of our study.

Geographical complexity and transportation difficulties were identified as barriers to the PNC service utilization in our study and the findings echo with the previous studies [49, 51]. Postnatal mothers in our study faced financial difficulties related to transportation costs, service expenses, and the costs associated with managing food and clothing. Additionally, our study revealed a shortage of adequate health workforce to support the PNC home-visit program. Financial and human resource challenges were similarly reported as barriers to postnatal service utilization in studies conducted in Nepal, [52] Indonesia, [50, 53], and Rwanda [47]. Lack of awareness was another barrier identified in our study that hinders postnatal mothers from seeking and utilizing postnatal services. This finding is consistent with studies conducted in Nepal, [54] Indonesia, [50, 53] and Ethiopia [55]. Though not found significant during a quantitative analysis, disruption in PNC services due to the COVID-19 pandemic was identified during qualitative interviews in our study. The COVID-19 pandemic induced fear of transmission, transportation difficulties, discontinuity of routine services, re-orientation of services, and health workforce redirection towards the management of COVID-19 cases, among others, were some of the reasons for decreased maternal service utilization during the COVID-19 pandemic, as reported in different studies [45, 56–59]. These findings were also concluded by our study.

Mothers rarely experience peaceful days after giving birth as they have to tackle the task of balancing infant care and self-care right away. They are frequently confronted with a combination of infant crying and personal tiredness. Our study found that visiting health facilities in the early days of postpartum was challenging due to the physical weakness of postnatal mothers, and the study conducted in Switzerland supports these findings [60]. Decision-making power was found to be mainly with in-laws, and they often restricted mothers from visiting health facilities in our study. Difficulty in convincing elder in-laws was also found in a study conducted in India [61].

## 5. Strengths and limitations

This is a mixed-method study that collected information from multiple sources to depict the situation of PNC service utilization in the Pyuthan district. The mixing of datasets has provided a better understanding of the problem and yielded rich information. Furthermore, the triangulation of the data has strengthened the findings of our study. The probability sampling method has been used to select samples, allowing for findings to be generalized. Moreover, weighted analysis has been carried out to provide unbiased results. Measures such as verification of the transcribed and translated scripts, member checking, inter-coder agreement, and process documentation were employed to maintain the rigor of the study.

The Pyuthan district was purposively selected, limiting the generalizability of the findings outside the district. Furthermore, due to time and resource constraints, the researcher was only able to include one rural municipality out of seven, which limits the representation of the rural stratum Postnatal mothers whose numbers could not be obtained from health facility and FCHVs records were excluded from the sampling frame, which may affect the true representation of the population. Although certain criteria were set to select samples for qualitative interviews, they did not capture all the socio-cultural, economic, and ethnic diversity of the district.

## 6. Conclusion

The utilization of PNC services as per protocol is relatively low (38.43%) in the district compared to other maternal health service indicators. Knowledge about PNC was identified as an independent predictor PNC service utilization as per protocol. Therefore, creating awareness among postnatal mothers about PNC check-ups, services, and danger signs through facility and community-based programs is crucial. Additionally, being delivered in a birthing center increases the likelihood of utilizing PNC services as per protocol, emphasizing the need to enhance institutional deliveries.

Despite decades-long efforts to improve maternal and neonatal health through various policies, from the Safe Motherhood Policy of 1998 to recent advancements such as the Reproductive Health Rights Act of 2018, the Continuum of Care Guideline of 2019, and the Nepal Safe Motherhood and Newborn Health Road Map 2030, progress in the postnatal component has been relatively slow compared to other aspects of maternal health care. Our study has observed that initiatives taken by local governments were more effective than those taken centrally in improving maternal health services. Therefore, revision of the policies, aligning with the federal structure and providing space for contextualized intervention, is needed. Furthermore, our study draws the attention of policymakers to the importance of using an equity lens when formulating policies and programs, as service utilization was found to be significantly impacted by broader socio-economic and geographical determinants. It is high time to revise the universal nature of the safe motherhood program and make it more targeted toward those in need. Although notable progress in PNC service utilization was observed in districts where the PNC home visit program was implemented, and our study identified it as a significant facilitator of service utilization, challenges such as inadequate financing and insufficient health workforce remain. The Family Welfare Division, while emphasizing extensive coverage of the PNC home visit program, should also address the issues of human resources and financing to ensure the program's sustainability. Collaboration with local levels is imperative to design more localized and tailored programs. Concerned stakeholders at federal, provincial, and local levels can leverage evidence from our study to develop strategic approaches and advocate for improved PNC service utilization, particularly in Pyuthan district and similar contexts more broadly.

Nepal, as one of the 81 countdown countries with relatively poor progress on reproductive, maternal, and child health indicators, particularly PNC [7] should focus on improving access to quality services. Establishing and operationalizing basic hospitals (5–15 beds) at each local level, a transformative project for the health sector outlined in the fifteenth plan [62], could enhance the reach of quality maternal health services to the community. Although our mixed-method study provided a deeper understanding of the factors associated with PNC service utilization, including its facilitators and barriers, information on the quality of PNC services remains insufficient. Therefore, further studies on the quality of PNC services and evaluations of the PNC home-visit program are recommended.

## Supporting information

**S1 Dataset.**
(XLSX)

**S2 Dataset. Coding framework.**
(XLSX)

**S1 File. Reflexivity.**
(DOCX)

## Acknowledgments

Authors would like to thank the School of Public Health, Patan Academy of Health Sciences, Pyuthan Health Office, Pyuthan Municipality, Jhimruk Rural Municipality and all the participants.

## Author Contributions

**Conceptualization:** Tulsi Ram Thapa, Reshu Agrawal Sagtani, Anita Mahotra, Ravi Kanta Mishra, Saraswati Sharma, Sudarshan Paudel.

**Data curation:** Tulsi Ram Thapa.

**Formal analysis:** Tulsi Ram Thapa, Reshu Agrawal Sagtani, Sudarshan Paudel.

**Methodology:** Tulsi Ram Thapa, Reshu Agrawal Sagtani, Anita Mahotra, Sudarshan Paudel.

**Resources:** Tulsi Ram Thapa.

**Software:** Tulsi Ram Thapa.

**Supervision:** Reshu Agrawal Sagtani, Ravi Kanta Mishra, Sudarshan Paudel.

**Validation:** Tulsi Ram Thapa, Reshu Agrawal Sagtani, Anita Mahotra, Ravi Kanta Mishra, Saraswati Sharma, Sudarshan Paudel.

**Visualization:** Tulsi Ram Thapa.

**Writing – original draft:** Tulsi Ram Thapa.

**Writing – review & editing:** Tulsi Ram Thapa, Reshu Agrawal Sagtani, Anita Mahotra, Ravi Kanta Mishra, Saraswati Sharma, Sudarshan Paudel.

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
