## [Decision Letter · Decision Letter 0]

6 May 2024

PONE-D-24-09868Factors affecting postnatal care service utilization in Pyuthan district: A mixed method studyPLOS ONE

Dear Dr. Thapa,

Thank you for submitting your manuscript to PLOS ONE. After careful consideration, we feel that it has merit but does not fully meet PLOS ONE’s publication criteria as it currently stands. Therefore, we invite you to submit a revised version of the manuscript that addresses the points raised during the review process.

Please address the comments from all reviewers.  

We look forward to receiving your revised manuscript.

Kind regards,

Kshitij Karki, MPH, MA

Academic Editor

PLOS ONE

Journal Requirements:

2. In the ethics statement in the Methods, you have specified that verbal consent was obtained. Please provide additional details regarding how this consent was documented and witnessed, and state whether this was approved by the IRB

"The authors declare that they have no competing interests."

Additional Editor Comments:

I have not seen the description of ‘As per protocol’. Please provide 1-2 sentences relevant to the subject matter relating to protocol.

Table 7, How did you measure Overall accessibility and Overall Satisfaction? Please explain (either in data analysis or in explanation during

In Table 8, please remove Pyuthan District and make it total – How do the two municipalities represent the entire Pyuthan District?

Please explain more about multivariable logistic regression and adjustment.

What new value is provided by your research in the sector for policy, program and research?

Reviewers' comments:

Reviewer's Responses to Questions

**Comments to the Author**

1. Is the manuscript technically sound, and do the data support the conclusions?

Reviewer #1: Partly

Reviewer #2: Yes

2. Has the statistical analysis been performed appropriately and rigorously? 

Reviewer #1: I Don't Know

Reviewer #2: Yes

3. Have the authors made all data underlying the findings in their manuscript fully available?

Reviewer #1: Yes

Reviewer #2: Yes

4. Is the manuscript presented in an intelligible fashion and written in standard English?

Reviewer #1: Yes

Reviewer #2: Yes

5. Review Comments to the Author

Reviewer #1: Maternal and child death is very painful. It is because of many reasons one is most of the causes of their death are preventable. Even if there is a poor trend of using the revealed evidence for planning, and acting on the research recommendations, doing such research has its own share.

As you said your method as well as the result might not be unique.

I'd have doubts about how much to take care of to avoid selection bias, most of the participants with a smooth delivery history.

In your discussion part, you used only one reference I don't know why you did this. Do you think that PNC home visits and PNC facility visits have the same recommendations?

Reviewer #2: Factors affecting postnatal care service utilization in Pyuthan district: A mixed method study

Introduction

1.I suggest that add the current situation of postnatal care globally in the second paragraph of the Introduction , rather than only describing the current situation in Nepal only.

2. There are few description in the status of postpartum care in Pyuthan.I suggest that describe the problems in the utilization of postnatal care services in Pyuthan through data.Then the purpose of this study is described.

Materials and method

1.Why do you choose Jhimruk Rural Municipality and Pyuthan Urban Municipality of Pyuthan district for research.

2.The interview outline of the qualitative survey or the main framework of the interview were not described.I suggest adding framework.

3.I suggest that the mixed method is specifically described, as it appears in the title and may be important.

Results

1.In the analysis of qualitative results, I suggest that attention paid to the analysis of the reason of certain phenomena, for example, 318 PNC home visits only described the current situation, before leaving the hospital, only the mother who gave birth in the health care institution received the first PNC examination. No follow-up tests were performed on the third and seventh days. So why wasn't there a follow-up examination? The interviews could have been semi-open. The purpose of finding the real reasons were and how to fix them. In line 326 Incentives, the same problem is suggested to be added.

6. PLOS authors have the option to publish the peer review history of their article (what does this mean?). If published, this will include your full peer review and any attached files.

Reviewer #1: No

Reviewer #2: No

---

## [Author Response · Author response to Decision Letter 0]

20 Jun 2024

Editors Comments

• Thank you so much for your suggestions. We have developed the manuscript as per the PLOS ONE’S style requirement and listed the supporting information files that are attached with the manuscript at the end of manuscript with the heading “Supporting Information”.

2. In the ethics statement in the Methods, you have specified that verbal consent was obtained. Please provide additional details regarding how this consent was documented and witnessed, and state whether this was approved by the IRB

• We appreciate your concern regarding the ethics statement. In Nepal, the first wave of COVID-19 lasted from August 2020 to January 2021, while the second wave (Delta variant) occurred between April and October 2021. Considering the risk of COVID-19 transmission, the Institutional Review Committee of Patan Academy of Health Sciences approved conducting interviews via telephone, obtaining verbal consent from each participant. All participants were briefed on the study's objectives, and their voluntary participation was ensured. They were encouraged to ask questions about the study and were informed that they had the right to withdraw from the interview at any time without providing any explanation. Participants were explicitly informed that there were no direct benefits or harms associated with participating in this study. Verbal informed consent was obtained before starting the interview. The entire process, from explaining the study’s objectives to obtaining verbal consent, was audio recorded for each participant. A unique ID was assigned while keeping the record of each verbal consent. The detailed explanation of ethics statement has now been incorporated into the methodology section under the sub-heading “Ethical consideration”.

"The authors declare that they have no competing interests."

• Thank you so much for your advice. We have now included the statement, “The authors have declared that no competing interests exist”, in both the online submission form and the cover letter.

• Thank you so much for your guidance. We have now included the statement, “All relevant data are within the manuscript and its supporting information files”, in the Data Availability Statement on the submission form. All supporting information files are attached along with the manuscript.

5. Please include captions for your Supporting Information files at the end of your manuscript, and update any in-text citations to match accordingly. Please see our Supporting Information guidelines for more information: http://journals.plos.org/plosone/s/supporting-information

• Thank you so much for your feedback. We have included the captions for the supporting information files at the end of the manuscript and updated the in-text citations, accordingly, following the supporting information guidelines of PLOS ONE.

Additional Editor Comments:

I have not seen the description of ‘As per protocol’. Please provide 1-2 sentences relevant to the subject matter relating to protocol

• Thank you so much for your suggestion. We have added the definition of ‘PNC service utilization as per protocol’ in the methodology section under the subheading “Operational Definitions”.

Table 7, How did you measure Overall accessibility and Overall Satisfaction? Please explain (either in data analysis or in explanation during

• We appreciate your concern about the measurement of overall accessibility and overall satisfaction in our study. We calculated the composite scores for overall accessibility and overall satisfaction, which were then dichotomized based on the criteria mentioned in the “Operational Definitions” section. The criteria for overall accessibility and overall satisfaction are explained in the operational definition and now we have added this in the description of Table 7 too.

In Table 8, please remove Pyuthan District and make it total – How do the two municipalities represent the entire Pyuthan District?

• We sincerely appreciate your thoughtful concern regarding the generalizability of the study findings to the entire Pyuthan District. However, efforts have been made to make the study findings as generalizable to the district as possible. A probability sampling method was employed at each stage of the sampling process to ensure the findings could be inferred to the larger population. The district was first divided into two strata (urban and rural), and one local level was selected randomly from each stratum. Additionally, the probability proportionate to size (PPS) method was used to select clusters (health facilities) from each stratum, and systematic random sampling was used to select study participants from each cluster. Weighted analysis was carried out to produce an unbiased estimate. Due to resource and time constraints, we were unable to include more rural municipalities from the district. Considering this limitation and in alignment with the research questions and title of the study, we would be thankful if you could allow us to use 'Pyuthan district' instead of 'total.' In acknowledgment of your concern, we have included this in our limitations section. If you still feel that we cannot infer at the district level, we will revise it as per your recommendation. 

Please explain more about multivariable logistic regression and adjustment.

• Thank you so much for your feedback. We have added an explanation about multivariable logistic regression and adjustment in subheadings '3.7.2' and '3.7.3' of the “Results” section.

What new value is provided by your research in the sector for policy, program and research?

• We appreciate your feedback to add what new value our research provides in the health sector in terms of policy, programs and research. Unlike many studies that focus on identifying barriers to postnatal care, our mixed method study aims to identify both facilitators and barriers, providing a more comprehensive understanding of the context-specific factors. The study identifies specific socioeconomic, demographic, and accessibility factors influencing PNC service utilization. Key predictors include socioeconomic status, place of delivery, knowledge of postnatal care, and access to motorable roads. The research design allowed us to triangulate and validate the findings of quantitative and qualitative study. Our study has observed that initiatives taken by local governments were more effective than those taken centrally in improving maternal health services. Therefore, revision of the policies, aligning with the federal structure and providing space for contextualized intervention, is needed. Furthermore, our study draws the attention of policymakers to the importance of using an equity lens when formulating policies and programs, as service utilization was found to be significantly impacted by broader socio-economic and geographical determinants. The detailed description of new value adds by our research along with conclusion and recommendation have been explained in the “Conclusion” section.

Review Comments to the Author

Reviewer #1:

Maternal and child death is very painful. It is because of many reasons one is most of the causes of their death are preventable. Even if there is a poor trend of using the revealed evidence for planning, and acting on the research recommendations, doing such research has its own share. 

As you said your method as well as the result might not be unique.

I'd have doubts about how much to take care of to avoid selection bias, most of the participants with a smooth delivery history.

• We truly agree with your perspective on maternal and child death and understand that the gravity of the pain incurred by their deaths is unbearable. The inability to prevent preventable maternal and child deaths deeply troubles us as responsible citizens of the country. Nepal is far behind in achieving the SDG target of reducing the MMR to below 70 per hundred thousand live births and ending preventable maternal and child deaths.

Though much of the research evidence is not fully translated into policy action, such evidence always plays an important role in rationalizing advocacy for policy and program design.

To avoid selection bias, a probability sampling method was employed at every stage of the sampling process to ensure that every individual in the sampling frame had a probability of being selected. The district was first divided into two strata (urban and rural), and one local level was selected randomly from each stratum. Additionally, the probability proportionate to size (PPS) method was used to select clusters (health facilities) from each stratum, and systematic random sampling was used to select study participants from each cluster. Efforts were made to make the sampling frame as representative as possible by using both health facility records and corresponding FCHV registers.

Regarding smooth delivery history, if we look at the delivery trend in Pyuthan District (http://dohs.gov.np/data-fact-sheet/) from FY 2016/17 to 2019/20, the proportion of deliveries by cesarean section (C/S) ranged between 4.7% and 7.5%. In our study, the proportion of C/S deliveries was observed to be 8.15%, and four (1.64%) postnatal mothers had assisted vaginal deliveries. Considering this evidence, we believe that our samples adequately captured various types of deliveries. However, postnatal mothers who were not registered at health facilities or in FCHV registers were excluded from the study, and this limitation has already been mentioned in the limitations section.

In your discussion part, you used only one reference I don't know why you did this. Do you think that PNC home visits and PNC facility visits have the same recommendations?

• We appreciate your careful feedback. However, while revisiting the discussion section of our manuscript, we observed that we have used several references in this section to compare and contrast our research findings and present our critical perspective. Only one reference might have been visible due to technical errors. The revised manuscript will not have this problem. If issues persist, please let us know, and we will resolve them accordingly.

We agree that PNC home visits and PNC at health facilities should not have the same recommendations. For PNC home visits, the issue of inadequate funding and health workforce should be addressed for its sustainable and smooth operation. While the quality of PNC services provided at health facilities should be enhanced to improve PNC service utilization. Detailed recommendations have been explained in the “Conclusion” section.

Reviewer #2:

Introduction 

1. I suggest that add the current situation of postnatal care globally in the second paragraph of the Introduction, rather than only describing the current situation in Nepal only.

2. There are few description in the status of postpartum care in Pyuthan. I suggest that describe the problems in the utilization of postnatal care services in Pyuthan through data. Then the purpose of this study is described.

• Thank you so much for your insightful feedback. We have now included the current situation of postnatal care globally and problems of PNC service utilization in Pyuthan district in the second paragraph of “Introduction” section.

Materials and method

1. Why do you choose Jhimruk Rural Municipality and Pyuthan Urban Municipality of Pyuthan district for research.

• We appreciate your concern regarding selection of study sites. The Pyuthan district was first divided into two strata (urban and rural). Jhimruk Rural Minicipality from rural stratum and Pyuthan Urban Municipality from urban stratum were selected randomly. 

2. The interview outline of the qualitative survey or the main framework of the interview were not described. I suggest adding framework.

• We appreciate your thoughtful feedback regarding the use of a specific framework to guide the qualitative study. Since we employed a mixed-method design with quantitative dominance (QUAN+qual), no specific framework was used for the qualitative portion. However, methodological robustness has been maintained throughout the qualitative study to enhance its rigor.

A separate interview guide was used for in-depth interviews and key informant interviews. Braun and Clarke’s six-step thematic analysis approach (familiarizing with the data, generating initial codes, generating initial themes, reviewing themes, naming and defining themes, and producing the report) was used for qualitative data analysis. Codes were generated by two independent coders to ensure inter-coder agreement (ICA) analysis. A coding framework was developed based on two predetermined themes: facilitators and barriers. Codes for other themes, such as the maternal health service context and knowledge of PNC, were identified inductively. The details of the coding and theme generation have been attached as supporting information files. Additionally, the researcher’s reflexivity has also been included in supporting information file that reflects upon the trustworthiness of the qualitative research.

Results

In the analysis of qualitative results, I suggest that attention paid to the analysis of the reason of certain phenomena, for example, 318 PNC home visits only described the current situation, before leaving the hospital, only the mother who gave birth in the health care institution received the first PNC examination. No follow-up tests were performed on the third and seventh days. So why wasn't there a follow-up examination? The interviews could have been semi-open. The purpose of finding the real reasons were and how to fix them. In line 326 Incentives, the same problem is suggested to be added.

• Thank you so much for your suggestions. We used semi-structured, open-ended interview guidelines for in-depth and key-informant interviews in our study. Probing was done whenever further information or clarification was needed from the participants. Although we encountered some difficulties with ice-breaking and maintaining the flow of the interview in a few cases, efforts were made to make the discussions livelier and to obtain richer information.

We now tried to address the issues raised above regarding PNC home visits and incentives by explaining them under the respective themes (PNC home visit, Incentives) of the qualitative results, with verbatim quotes to support these explanations.

---

## [Decision Letter · Decision Letter 1]

11 Jul 2024

Factors affecting postnatal care service utilization in Pyuthan district: A mixed method study

PONE-D-24-09868R1

Dear Tulsi Ram Thapa,

We’re pleased to inform you that your manuscript has been judged scientifically suitable for publication and will be formally accepted for publication once it meets all outstanding technical requirements.

Kind regards,

Kshitij Karki, MPH, MA

Academic Editor

PLOS ONE

Additional Editor Comments (optional):

Reviewers' comments:

Reviewer's Responses to Questions

**Comments to the Author**

1. If the authors have adequately addressed your comments raised in a previous round of review and you feel that this manuscript is now acceptable for publication, you may indicate that here to bypass the “Comments to the Author” section, enter your conflict of interest statement in the “Confidential to Editor” section, and submit your "Accept" recommendation.

Reviewer #2: All comments have been addressed

2. Is the manuscript technically sound, and do the data support the conclusions?

Reviewer #2: Yes

3. Has the statistical analysis been performed appropriately and rigorously? 

Reviewer #2: Yes

4. Have the authors made all data underlying the findings in their manuscript fully available?

Reviewer #2: Yes

5. Is the manuscript presented in an intelligible fashion and written in standard English?

Reviewer #2: Yes

6. Review Comments to the Author

Reviewer #2: Thanks for the letter from the editor. I have carefully reviewed the author's reply and revised manuscript, and found that the author responded to every suggestion very seriously, and there are no other comments.

7. PLOS authors have the option to publish the peer review history of their article (what does this mean?). If published, this will include your full peer review and any attached files.

Reviewer #2: No

---

## [Editor Report · Acceptance letter]

20 Jul 2024

PONE-D-24-09868R1 

PLOS ONE

Dear Dr. Thapa, 

I'm pleased to inform you that your manuscript has been deemed suitable for publication in PLOS ONE. Congratulations! Your manuscript is now being handed over to our production team.

Kind regards, 

on behalf of

Dr. Kshitij Karki 

Academic Editor

PLOS ONE